# Graphitic Carbon Nitride/MOFs Hybrid Composite as Highly Selective and Sensitive Electrodes for Calcium Ion Detection

**DOI:** 10.3390/molecules28248149

**Published:** 2023-12-18

**Authors:** Ahmed S. Abou-Elyazed, Shilin Li, Gehad G. Mohamed, Xiaolin Li, Jing Meng, Safa S. EL-Sanafery

**Affiliations:** 1Institute of Intelligent Manufacturing Technology, Shenzhen Polytechnic University, Shenzhen 518055, China; ahmedphysical90@gmail.com (A.S.A.-E.); islishilin@163.com (S.L.); 2Chemistry Department, Faculty of Science, Menoufia University, Shebin El-Kom 32512, Egypt; safasaid34@yahoo.com; 3Heilongjiang Province Key Laboratory of Laser Spectroscopy Technology and Application, Harbin University of Science and Technology, Harbin 150080, China; 4Chemistry Department, Faculty of Science, Cairo University, Giza 12613, Egypt; ggenidymohamed@sci.cu.edu.eg; 5Nanoscience Department, Basic and Applied Sciences Institute, Egypt-Japan University of Science and Technology, New Borg El Arab 21934, Egypt; 6School of Civil and Environmental Engineering, Harbin Institute of Technology (Shenzhen), Shenzhen 518055, China

**Keywords:** metal-organic framework, graphitic carbon nitride, carbon-paste electrodes, HPLC, SSM, MPM

## Abstract

The metal–organic framework (MOF) is a class of materials that exhibits a notable capacity for electron transfer. This unique framework design offers potential applications in various fields, including catalysis, gas storage, and sensing. Herein, we focused on a specific type of MOF called Ti-MOF. To enhance its properties and functionality, the composite material was prepared by incorporating graphitic carbon nitride (g-C_3_N_4_) into the Ti-MOF structure. This composite, known as g-C_3_N_4_@Ti-MOF, was selected as the active material for ion detection, specifically targeting calcium ions (Ca^2+^). To gain a comprehensive understanding of the structural and chemical properties of the g-C_3_N_4_@Ti-MOF composite, several analytical techniques were employed to characterize the prepared g-C_3_N_4_@Ti-MOF composite, including X-ray diffraction (XRD), SEM-EDX, and FT-IR. For comparison, different pastes were prepared by mixing Ti-MOF or g-C_3_N_4_@Ti-MOF, graphite, and *o*-NPOE as a plasticizer. The divalent Nernstian responses of the two best electrodes, I and II, were 28.15 ± 0.47 and 29.80 ± 0.66 mV decade^−1^, respectively, with concentration ranges of 1 µM–1 mM and 0.1 µM–1 mM with a content 1.0 mg Ti-MOF: 250 mg graphite: 0.1 mL *o*-NPOE and 0.5 mg g-C_3_N_4_@Ti-MOF: 250 mg graphite: 0.1 mL *o*-NPOE, respectively. The electrodes showed high sensitivity and selectivity for Ca^2+^ ions over different species. The suggested electrodes have been successfully employed for Ca^2+^ ion measurement in various real samples with excellent precision (RSD = 0.74–1.30%) and accuracy (recovery = 98.5–100.2%), and they exhibited good agreement with the HPLC.

## 1. Introduction

The metal–organic frameworks (MOFs) family of porous materials has garnered significant interest in recent times because of its distinctive structural, chemical, and physical characteristics [1,2]. MOFs are made up of organic ligands joining metal ions or clusters to produce a highly organized, porous crystalline structure [3,4]. The tunable nature of MOFs allows for the creation of a wide range of materials, with varying pore sizes, surface areas, and selectivity towards gases or molecules [1]. Among the best potential uses of MOFs is the development of sensors. MOFs have shown great potential in detecting a variety of analytes, such as gases, vapors, minerals, and even biological molecules [3]. The high surface area and tunable pore size of MOFs allow for the selective adsorption of specific analytes, leading to highly sensitive and selective sensors [3,5].

MOFs have been used in gas sensing applications, such as detecting toxic gases, like CO_2_ and NO_2_, and volatile organic compounds (VOCs), which are commonly found in the environment [6]. Furthermore, MOFs have shown potential in biological sensing applications, such as detecting proteins and DNA. In addition, MOFs can be used as transducers in electrochemical and optical sensing, where changes in the structure of MOFs upon analyte adsorption can be detected through changes in conductivity, fluorescence, or other optical properties [7]. The development of MOF-based sensors has the potential to revolutionize a wide range of fields, from environmental monitoring to medical diagnostics [8]. MOFs offer advantages over traditional sensing materials, including high sensitivity, selectivity, and reversibility. Moreover, the ability to tune the properties of MOFs makes them highly adaptable to a variety of sensing applications [9].

In particular, MOFs have shown great potential in the detection of calcium ions (Ca^2+^), which are essential in numerous physiological processes in living organisms [10]. Calcium ions are crucial in a wide range of biological processes, including muscle contraction, neurotransmission, and regulation of enzyme activity [11]. Therefore, the detection of Ca^2+^ ions is of great importance in medical diagnostics, environmental monitoring, and the food industry. Traditional methods for Ca^2+^ ion detection is highly sensitive but require specialized equipment and are time-consuming. Therefore, there is a need for the development of alternative, more efficient, and cost-effective methods for Ca^2+^ ion detection. MOFs offer an attractive solution for the detection of Ca^2+^ ions due to their high selectivity, sensitivity, and reversible adsorption properties. The tunable nature of MOFs allows for the design of materials with specific binding sites that can selectively adsorb Ca^2+^ ions over other cations [12]. 

Furthermore, MOFs can be designed to exhibit fluorescence or colorimetric changes upon Ca^2+^ ion binding, allowing for simple and rapid detection without the need for complex equipment [13]. Several MOFs have been reported for the detection of Ca^2+^ ions, including UiO-66, MIL-101, and ZIF-8. For example, UiO-66 has been reported to exhibit remarkable selectivity towards Ca^2+^ ions over other cations, such as magnesium and sodium ions [14]. Similarly, MIL-101 has shown high sensitivity towards Ca^2+^ ions, with a detection limit as low as 10 nM. On the other hand, ZIF-8 has been reported to exhibit reversible adsorption of Ca^2+^ ions, allowing for the reuse of the material [15]. The Ca^2+^ ion is determined using many different techniques like high performance liquid chromatography (HPLC) [16,17] and atomic absorption spectrometry [18,19]. These techniques need highly skilled personnel, high instrument and operational expenses, preparatory sample treatment that raises the possibility of sample contamination, and challenging instrument setup. Utilizing these techniques for in-field, real-time measurements is challenging due to all these factors [20]. 

In this work, ion-selective sensors are made from a g-C_3_N_4_@Ti-MOF composite, which is used to detect the concentration of Ca^2+^ ions in the analytes. The Ti-MOF and graphitic carbon nitride (g-C_3_N_4_@Ti-MOF) composite were prepared using the facile method and utilized as active materials to determine Ca^2+^ ions in many real samples using carbon-paste ion-selective electrodes. The structure of the prepared g-C_3_N_4_@Ti-MOF composite was characterized using scanning electron microscopy (SEM) and X-ray diffraction, BET, and Fourier transform infrared spectroscopy (FT-IR). Additionally, the response time, the impact of temperature, and pH were examined for the proposed electrodes (I) and (II). 

## 2. Experimental Methods

### 2.1. Instrumentation and Chemicals

Anhydrous titanium isopropoxide (Ti_4_(OCH_3_)_16_), 1,4-benzenedicarboxylic acid (BDC), methanol, dimethylforamide (DMF), urea, acetone, and Aldrich provided the synthetic 1–2 µm graphite powder. Fluka provided the o-Nitrophenyloctyl Ether (*o*-NPOE). BDH provided tricresylphosphate (TCP), dibutyl phthalate (DBP), and dioctyl phthalate (DOP). El Nasr Company provided all the salts that were utilized as materials for interference. Baby milk powder (Bebelac 2) was supplied from Danon company. The CAL-MAG pharmaceutical drug was supplied from Hochster pharmaceutical industries, and deionized water. 

### 2.2. Synthesis of Titanium-Based MOFs

MIL-125 was obtained starting from terephthalic acid (332 mg), and titanium isoproproxide (0.6 mL) should be introduced in a solution of dimethylformamide and dry methanol (1:1 *v*/*v*). The mixture was stirred gently for 5 min at room temperature and then further introduced in a Teflon line autoclave at a specific temperature (150 °C) for 15 h. The white solid was recovered via centrifugation and washed twice with acetone. It was then dried in an oven at 80 °C for 24 h and then dried under vacuum at 150 °C for 24 h as reported in the literature [21].

### 2.3. g-C_3_N_4_ Synthesis

Using a horizontal furnace and an alumina crucible, graphitic carbon nitride (g-C_3_N_4_) was produced by thermally treating urea-filled air up to 500 °C for 2 h at a heating rate of 10 °C min^−1^ [22].

### 2.4. Graphitic Carbon Nitride/Ti-MOF Composite Synthesis

A certain amount of graphitic carbon nitride was weighed (10 wt.% based the weight of Ti-MOF), and it was ground with Ti-MOF support in a mortar for 10 min at room temperature. After that, the composite was separated from the mortar and activated under vacuum for 24 h at 150 °C. The pure MOF was referred to as MIL-125(Ti), and its composite was referred to as g-C_3_N_4_@Ti-MOF.

### 2.5. Preparation of Carbon-Paste Electrodes 

Carbon-paste electrodes (CPEs) were made by thoroughly combining MOF, which varied in mass from 0.20–2.0 mg, with graphite powder 250 mg and various 0.10 mL plasticizers like DOP, TCP, *o*-NPOE, DOS, or DBP. By carefully combining the ingredients in an agate pestle and mortar and then vigorously pressing the pestle on the work surface, a very fine homogeneity was created. After packing the ready-made paste into the electrode body’s hole, the carbon paste was spread out onto a wet filter paper and polished until it appeared shiny. The electrode body was a 12 cm long Teflon holder with a hole for retaining the carbon paste that was 7 mm in diameter and 3.5 mm deep at one end. A stainless-steel rod was put into the middle of the holder to make electrical contact. To compress the paste and refresh the electrode surface as needed, the rod was movable and could be turned up or down. Excess paste was squeezed out and cleaned away to create a clean surface. A shining surface was then polished for confirmation on a sheet of paper.

### 2.6. Determination of Ca^2+^ in Baby Powder Milk Sample 

Amounts of 0.1 g of milk samples were placed in a 50 mL digestion flask, where they were mixed with 5 mL of concentrated HNO_3_ and heated on an electric hot plate at 800 °C for 2–3 h to obtain a clear, transparent digest. Then, the sample was heated on a heating plate to evaporate the excess acid, and a semidried mass was obtained. For the final determination, the solution was cooled to room temperature, then diluted to 25 mL with 0.2 mol L^−1^ nitric acid and filtered through 0.45 mL Whatman filter paper in a polyethylene flask. Finally, 100 milliliters of the sample solution were created by diluting the filtrate with deionized water.

### 2.7. Determination of Ca^2+^ Ions in the Pharmaceutical Drug

After finely powdering the CAL-MAG tablet and dissolving it in deionized water, the mixture was shaken up and then passed through filter paper and rinsed with more deionized water. A 100-milliliter measuring flask held the filtrate and washing mixture. The concentrated solution was serially diluted to create different concentrations. The potential of samples was measured using the reference electrode and the prepared sensors; the results were compared to the calibration plots of the drug solution, which served as the standard.

### 2.8. Selectivity Studies

This work used the separate solution method (SSM) and matched potential method (MPM) to analyze the selectivity coefficients of the electrode towards various cationic species of Ca^2+^. In SSM, the selectivity coefficient of prepared electrodes was evaluated by comparing the potential of two solutions of Ca^2+^ and interfering ion, and selectivity coefficients were determined using the following Equation (1) [23].
(1)$${\log K}_{A:B}{\;}^{\mathrm{SSM}}=\frac{E_{B}-E_{A}}{S}+(1-\frac{Z_{A}}{Z_{B}}){loga}_{A}$$
where *Z_B_* and *Z_A_* are the charge of the interfering ions and primary ion Ca^2+^; *S* represents the slope; *a_A_* represents the principal ion’s activity; and *E_A_* and *E_B_* represent the measured potentials of Ca^2+^ and the interfering ions, respectively. In MPM, the reference electrode and modified carbon-paste electrode were put in a beaker containing 10 µM of Ca^2+^ ion concentration as the reference solution, and then 0.1 mL of 100 mM of Ca^2+^ ion solution was added, and the potential was measured. Then, in the second beaker the potential of interference was measured, and 10 µM of Ca^2+^ of ion solution was added to the reference solution till the two potentials were the same, and from the following Equation (2) the selectivity coefficient can be determined [23].
*K*^*pot*^ _*A*, *B*_ = (*a*_*A*’_ − *a*_*A*_)/*a*_*B*_. (2)

### 2.9. Characterization of the Synthesized Ionophores

On a Rigaku (Tokyo, Japan) D/Max-2550 diffractometer outfitted with a SolX Detector-Cu K radiation with *λ* = 1.542 Å, XRD patterns were examined. Step scanning at *2θ* = 0.02° per second from 5° to 50° was used to collect data. Nitrogen sorption studies were carried out on a 3H-2000PS1 Gas Sorption and Porosimetry system at −196 °C. After degassing at 150 °C under vacuum for 2 h until the final pressure approached 110^−3^ Torr, the samples were periodically organized for analysis. Scanning electron microscope (SEM) images and elemental composition from EDX were monitored on a SUPRA 55 furnished with an acceleration voltage of 20 kV. Using a NicoLET iS10 spectrometer (Thermo Fisher Scientific, Waltham, MA, USA), Fourier transform infrared (FT-IR) spectroscopy was carried out. A Bruker (Billerica, MA, USA) Equinox 55 Fourier transform infrared spectrophotometer was used to record the spectra using the KBr pellet technique. Diffuse reflectance spectra were scanned with a resolution of 2 cm^−1^, resulting in a range of 500~4000 cm^−1^ for each measurement. Fabricated carbon-paste electrodes and a double-junction reference electrode made of silver-silver chloride (Metrohm (Herisau, Switzerland) 6.0222.100) were combined. A Jenway (Hong Kong, China) 3505 pH meter was used to measure pH. 

## 3. Results and Discussions

### 3.1. Structural Characterization

The synthesized composite of g-C_3_N_4_ with MIL-125(Ti) is referred to as g-C_3_N_4_@Ti-MOF. Practically, it contains 10% of g-C_3_N_4_ phase. Analyzing the X-ray diffraction (XRD) patterns is crucial for evaluating the synthesized materials’ results since it allows for the assessment of the MOF structural features and their maintenance in the presence of another phase (g-C_3_N_4_). Figure 1 displays the XRD patterns of the parent MOFs and the composites. The patterns of Ti-MOF follow those reported in the literature [24]. The preserved Ti-MOF structure was also found in the composite. However, the insertion of g-C_3_N_4_ resulted in a slight change to the lattice structure and shape, as seen by the diffraction peaks being less intense than those of the parent MOF. Moreover, two peaks at approximately 27.6 and 13.5 in the g-C_3_N_4_ X-ray diffraction patterns were linked to the inter-planar stacking of the graphitic layer [25].

SEM was used to examine the morphology of MIL-125(Ti), g-C_3_N_4_, and g-C_3_N_4_@Ti-MOF, as shown in Figure 2. The SEM images make it abundantly evident that g-C_3_N_4_ has wrinkled blossom morphology [26] (Figure 2a), and the particles of MIL-125(Ti) are irregular in shape (Figure 2b). However, in the instance of g-C_3_N_4_@Ti-MOF, the morphology has not been altered, but the particle size has been reduced (Figure 2c), leaving some porous bud shapes intact. The element present in g-C_3_N_4_@Ti-MOF was also determined using energy dispersive spectroscopy, as shown in Figure 2d. The existence and distribution of carbon, nitrogen, oxygen, and titanium in the composite material were confirmed in Figure 2d.

Moreover, FT-IR was used to investigate g-C_3_N_4_, Ti-MOF, and g-C_3_N_4_@Ti-MOF (Figure 3). The broad absorption band in the range of 3250–3500 cm^−1^ in the FT-IR spectrum of g-C_3_N_4_ indicated the presence of a N-H stretching vibration, whereas the absorption bands in the ranges 1420 and 1627 cm^−1^ correspond to the C-N and C=N stretching vibrations of the aromatic triazine ring system, respectively [27]. The fingerprint region at 1500 cm^−1^ in the FT-IR spectrum of Ti-MOF (Figure 3) is comparable to that reported in the literature [28]. The presence of dicarboxylate linker in the synthesized material is linked to the peaks at 1719 cm^−1^. The signal at 1409 cm^−1^ is associated with carboxylic acid O-C-O symmetric stretching [29]. Moreover, the Ti-O-Ti stretching vibration is linked to the peaks at 780, 680, and 535 cm^−1^ [30].

### 3.2. Effect of Electrode Composition

The kind and quantity of the ionophore, plasticizer, and amount of lipophilic additive used all have a major impact on the linear range, slope, and response time. Five different electrodes were prepared by changing the amount of the Ti-MOF ionophore from 0.20 to 2.0 mg since the amount of modifier in the paste is a crucial aspect for investigating the response of the electrode. The electrode with the composition 1.0 mg Ti-MOF, 250 mg graphite, and 0.10 mL TCP has the best Nernstian slope, measuring 25.15 ± 0.47 mV decade^−1^, according to Appendix A. The electrodes were found to be less stable at concentrations higher than this ratio, and the resulting slope of these concentrations was less like the slope of the divalent ion. This might be explained by the network oversaturation that prevented the complexation process and reduced sensitivity. 

Plasticizer, one of the main ingredients in the construction of an ion-selective electrode, improves the paste’s flexibility and softness, ensures the ionophore’s mobility, establishes the paste’s polarity, and gives the paste acceptable mechanical qualities. The effect of plasticizer was investigated since its chemical composition and polarity have a significant impact on the mobility of the ionophore, selectivity, and concentration range of ion-selective electrodes (ISEs). The most crucial qualities of plasticizers are high molecular weight, low vapor pressure, high lipophilicity, and great ability to dissolve all components of the sensor. The potentiometric response of electrodes manufactured with various plasticizers, such as DBP, *o*-NPOE, and DOP, was compared to those plasticized with TCP to investigate the effects of plasticizers on electrode performance. The data in Appendix A indicate that the electrode (I) that plasticized with *o*-NPOE performed better than the electrode that plasticized with TCP, and the working range concentration increased from 10 µM–1 mM to 1 µM–1 mM and gave a divalent Nernstian slope (28.15 ± 0.47 mV decade^−1^). Because of the high dielectric constant (ε = 24) and comparatively high molecular weight of *o*-NPOE, the electrode plasticized by *o*-NPOE demonstrated the highest sensitivity and a wider working concentration range. As explained before, the lipophilic additive has a major effect on the sensitivity of electrodes, so another ionophore g-C_3_N_4_@Ti-MOF was used in the preparation of electrodes instead of Ti-MOF. As shown in Appendix A, the use of g-C_3_N_4_@Ti-MOF gives higher sensitivity and wider linear range. Due to the presence of g-C_3_N_4_, which possesses electron-rich characteristics and H-bonding motifs because of the existence of N and H atoms, the electrode prepared using g-C_3_N_4_@Ti-MOF has better sensitivity than the electrode prepared with Ti-MOF. Furthermore, g-C_3_N_4_ reduced the particle size and created voids allowing the interaction of electrode active materials with calcium species, and it increased the adsorption characteristics of Ti-MOF. Additionally, Ca^2+^ ions are capable of intercalating into the interlayer spaces of g-C_3_N_4_. This intercalation process leads to introducing new electronic states in the interlayer region of g-C_3_N_4_. This intercalation can influence the transport properties and electronic conductivity of g-C_3_N_4_ [31,32]. As shown in Appendix A, electrode (II) with the composition 0.50 mg g-C_3_N_4_@Ti-MOF, 250 mg graphite, and 0.10 mL *o*-NPOE has the best Nernstian slope, measuring 29.15 ± 0.47 mV decade^−1^ and wider a concentration ranging from 0.1 µM–1 mM, and the mechanism of interaction of electrode (II) toward calcium ions is shown in Figure 4. 

### 3.3. Surface Charactrization

SEM was used to analyze the proposed electrodes’ morphology, as seen in Figure 5. After the proposed electrodes were prepared, they were immersed in a 10^−3^ mol L^−1^ Ca^2+^ solution for a duration of one hour. Both before and after soaking, the surfaces were inspected. Before soaking, the sensor’s surface is homogenous, porous, and full of cracks that help with Ca^2+^ ion diffusion, as seen in Figure 5. The spaces between the carbon grains filled after soaking due to the complex formation between the Ca^2+^ ions and the used donor atoms in the MOF.

### 3.4. Effect of Temperature

At 10, 25, 40, 50, 60, and 70 °C, calibration graphs were created to examine the thermal stability of the modified Ca^2+^ sensors. The standard cell potentials (*E°cell*) were plotted against (*t*-25) at each temperature, derived from the several calibration plots as the intercepts of those plots at [Ca^2+^] = zero (Figure 6). According to Antropov’s equation, a straight-line plot is produced, and as a result, the slope of this line corresponds to the electrodes’ respective isothermal coefficients, which are 1.33 and 1.50 mV/°C for electrodes (I) and (II), respectively [27]. The acquired isothermal coefficient values demonstrated a good thermal stability of the suggested electrodes within the applied temperature range. The electrodes exhibit good Nernstian behavior within the studied temperature range. As shown in Figure 6, electrode (I) was only stable to a temperature of 50 °C while electrode (II) was stable to a temperature of 60 °C. At higher temperatures, the forward reaction rate in which Ca^2+^ interacts with ionophore reduces, and the voltage created likewise does, which results in a reduction in the sensitivity of electrodes.

### 3.5. Response Time

The average period of time after immersing the projected electrodes in a series of Ca^2+^ solutions with concentration differences of ten times what the electrode needs to establish a steady potential response at intervals of 1 mV is called response time [33]. For both electrodes (I) and (II), concentration ranges of 1 µM to 1 mM and 0.1 µM to 1 mM were investigated for the response times of the prepared electrodes. The recommended sensors, as seen in Figure 7, respond incredibly quickly at 7.6 and 5 s for electrodes (I) and (II), respectively. The fact that the calibration only required a few minutes, as reported by the results, also confirms that the plasticizer’s and paste’s ingredients were the best available.

### 3.6. Effect of pH

At fixed concentration and temperature levels, the effect of pH on the electrode response was examined at the pH range of 2.0–14.0, and the relationship between the potential and the test solution’s pH (1 mM and 0.1 mM) was investigated. The potential readings and pH values were plotted, as seen in Figure 8. According to the findings, electrodes (I) and (II) were unaffected by pH variations in the range of 2.00–8.50. The electrodes become H^+^ sensitive at a pH lower than 2. This can be explained by the interference from H_3_O^+^, after which the potential values steadily rose, while the diminution at higher pH is most likely due to the precipitation of the calcium ion’s oxide or hydroxides.

### 3.7. Potentiometric Selectivity Evaluation

It investigated how the proposed electrodes responded when various foreign chemicals were present. In compliance with IUPAC guidelines, the matched potential method (MPM) and separate solutions method (SSM) were utilized to evaluate the potentiometric selectivity coefficients (*K_A,B_*). The results shown in Table 1 demonstrate that inorganic cation interference is minimal to nonexistent, indicating the high selectivity of the recommended electrodes for Ca^2+^. According to the hard–soft acid base hypothesis (HSAB) [23], the hardness of the ionophore and metal ions determines how well they interact with one other. The active sites where heavy metals and ionophores coordinate are N, S, and O and various atom binding sites such as O, N, S; N, O; N, S; O, S; etc. The soft binding sites in the ionophores are S and O, S, while the hard binding sites are N, O and O, N. The mechanism of sensing of the electrodes in the Ca^2+^ ion can be supposed to be based on the chelation between the oxygen and or nitrogen donor sites of the ionophores and the Ca^2+^ ions.

### 3.8. Reusability Study of Electrodes

By regularly making calibration graphs under ideal circumstances on various days and weeks, the lifetimes of the suggested sensors were examined. For the sensors under consideration, the electrodes have lifetimes of 14 and 17 weeks for electrodes (I) and (II), respectively, as illustrated in Figure 9. After this period, the electrodes’ sensitivity declines as the paste begins to lose its characteristics and possibly becomes saturated with Ca^2+^ ions. 

### 3.9. Analytical Application

As the prepared electrodes demonstrated no interference with multiple metals, the proposed electrodes were effectively employed to assess Ca^2+^ in genuine samples, such as baby milk powder and the pharmaceutical medication CAL-MAG, via direct, standard addition and calibration graph methods, as well as tap water samples. The Ca^2+^ ion was measured using HPLC according to the described procedure [19]. Table 2 illustrates the results of a comparison between the proposed electrodes and the HPLC method using the student’s *t*-test and variance ratio F-test. The results showed that the suggested electrodes were more efficient in terms of sensitivity and repeatability. The recovery percentage was found to be 98.33–100.0%, exhibiting the good accuracy of the suggested technique, and the computed RSD (%) values were less than 1.00%, suggesting the precision of the proposed approach.

## 4. Method Validation 

### 4.1. Inter- and Intra-Day Precision and Accuracy

On three distinct days, as well as on the same day, five repeat experiments with varied Ca^2+^ concentrations in powdered milk, CAL-MAG pharmaceutical samples, and pure form were conducted. The data gathered and shown in Appendix A demonstrate the validity, applicability, reproducibility, and repeatability of the recommended electrodes. The percent recovery for electrodes (I) and (II) is 99.30–99.90% and 98.35–100.0, respectively, demonstrating the great accuracy of the suggested procedure. The method’s reasonable repeatability was proved via its low relative standard deviation value.

### 4.2. LOQ and LOD [20]

The formulas used to calculate the LOQ and LOD were:LOD = 3SD/S, LOQ = 10SD/S
where (S) is the calibration curve’s slope, and (SD) is the intercept of the blank response’s standard deviation. The limit of detection (LOD) was estimated using the lowest Ca^2+^ concentration that could be readily detected. Elevated sensitivity is evident from the LOD values of electrodes (I) and (II), which are 1 and 0.1 µM, respectively. According to ICH Q2 (R1) recommendations (ICH, 2005), the lowest concentration at which a nonlinear calibration range was seen was used to determine the limit of quantification (LOQ). For electrodes (I) and (II), it was discovered to be 3.3 and 0.33 µM, respectively. The response properties for electrodes (I) and (II) are listed in Table 3.

### 4.3. Potentiometric Titration

Using both electrodes (I) and (II), the end point in the Ca^2+^ ion titration against standard EDTA solution has been identified. Potentiometric titration of 1 mM calcium ion solution versus 0.01 M EDTA solution was performed using the two recommended electrodes. The titration plot and end point were then established by plotting the potential measurements against the volume of titrant (EDTA) administered as shown in Figure 10.

### 4.4. Comparison Study

Table 4 compares the performance characteristic of the proposed sensors with previous ion-selective electrodes used for Ca^2+^ determination. The data in Table 4 reflect the dominance of the proposed MOF-based electrode over the other methods where the proposed electrode gave the widest concentration range, relatively lower detection limit, and lower RSD values than some of the other methods and relatively analogous to few cases. The RSD and the percentage recovery values reflect the successful use of the proposed sensor for the determination of the BF drug with high precision and accuracy. 

## 5. Conclusions

In this sense, the fabricated electrodes’ pastes were created by combining graphite, *o*-NPOE as a plasticizer, and either g-C_3_N_4_@Ti-MOF or Ti-MOF as a modifier. The proposed electrodes exhibited a very low detection limit and have good sensitivity, selectivity, and ease of use. The suggested approach is very cheap and simple to use. The suggested modified carbon-paste electrodes can test Ca^2+^ ions in a variety of samples without the use of complicated equipment or methods. They are used to measure Ca^2+^ ions using direct, calibration graph, and standard addition methods in real powdered milk samples and pharmaceutical drugs. The electrodes show very short response time and high thermal stability. Generally, direct potentiometry using g-C_3_N_4_@Ti-MOF ISEs has several advantages over other techniques, including low cost, low detection limits, and simplicity.

## Figures and Tables

**Figure 1 molecules-28-08149-f001:**
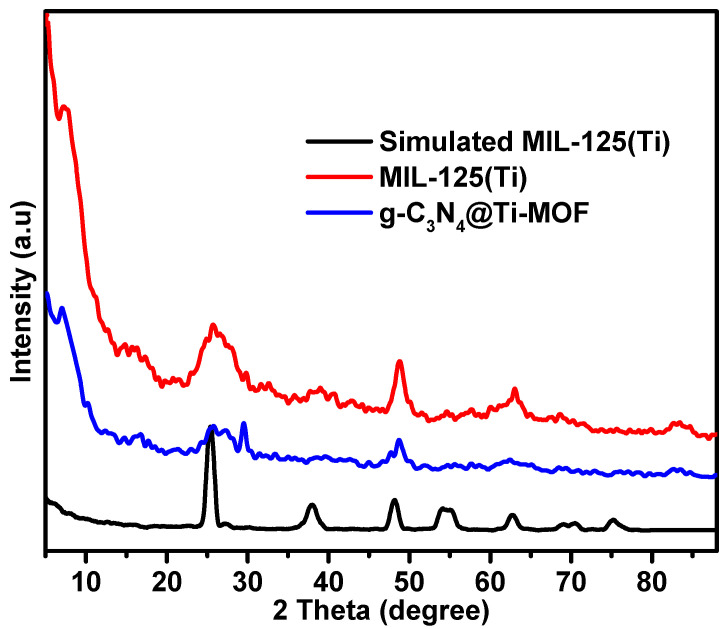
XRD patterns of MIL-125(Ti) and g-C_3_N_4_@Ti-MOF samples.

**Figure 2 molecules-28-08149-f002:**
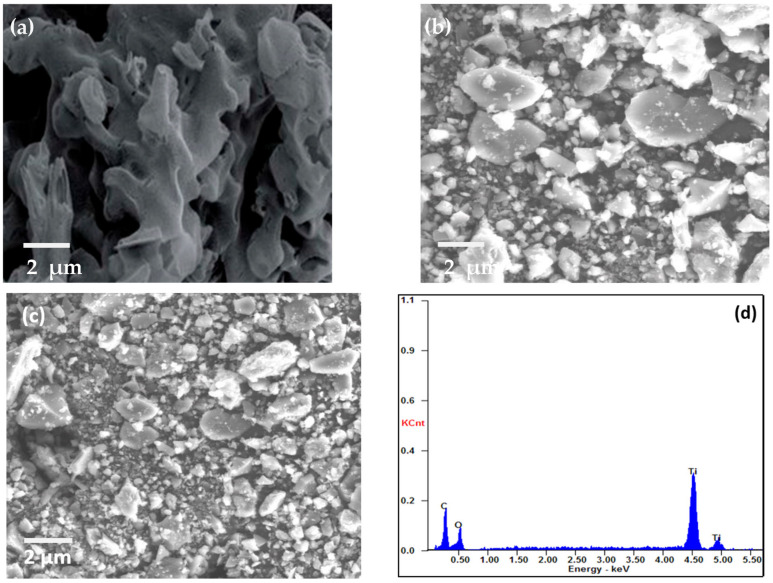
SEM images of g-C_3_N_4_ (**a**); MIL-125(Ti) (**b**); g-C_3_N_4_@Ti-MOF (**c**); and EDX of g-C_3_N_4_@Ti-MOF composite (**d**).

**Figure 3 molecules-28-08149-f003:**
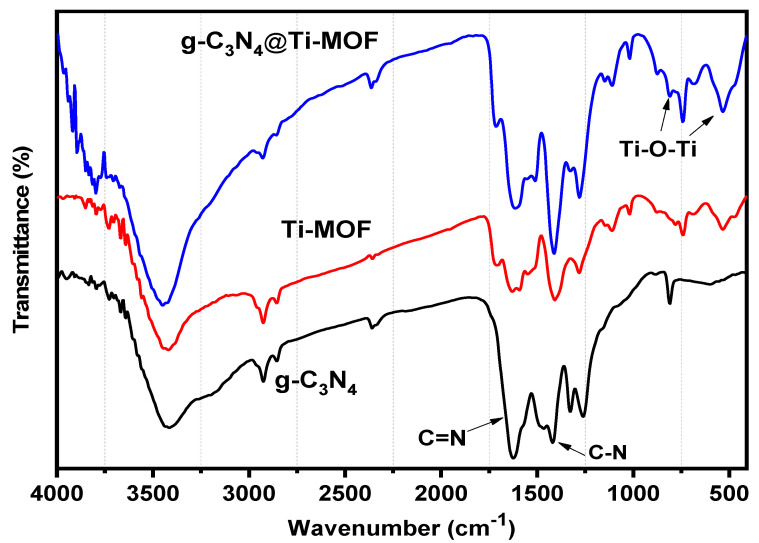
FT-IR spectra of g-C_3_N_4_ (black line), Ti-MOF (red line), and g-C_3_N_4_@Ti-MOF (blue line).

**Figure 4 molecules-28-08149-f004:**
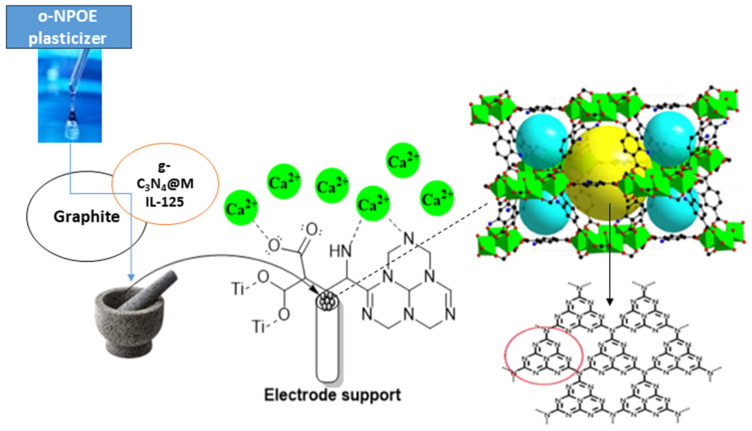
Schematic mechanism for the interaction of g-C_3_N_4_@Ti-MOF electrode toward Ca^2+^ ions (red cycle is referred to the triazine imide moiety in g-C_3_N_4_).

**Figure 5 molecules-28-08149-f005:**
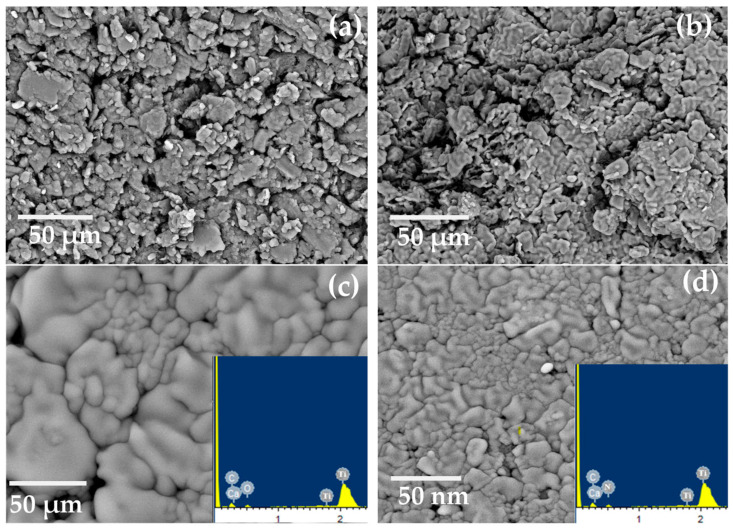
SEM images for electrodes (I) and (II) (**a**,**b**) before soaking and (**c**,**d**) after soaking in 10^−3^ mol L^−1^ of Ca^2+^ ions, and EDX after soaking (inset Figure).

**Figure 6 molecules-28-08149-f006:**
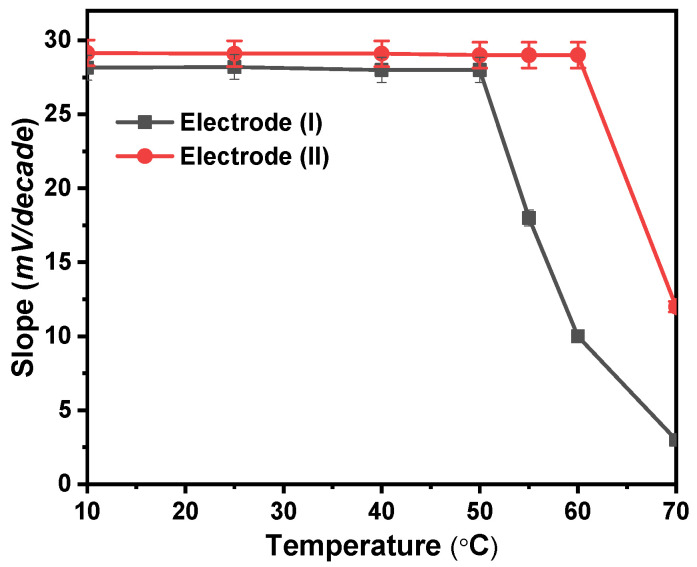
Temperature effect on the proposed electrodes (I) and (II).

**Figure 7 molecules-28-08149-f007:**
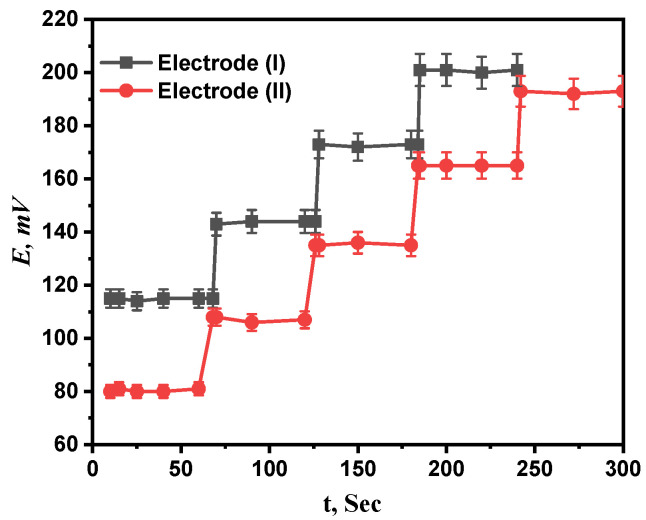
The response time of proposed electrodes (I) and (II).

**Figure 8 molecules-28-08149-f008:**
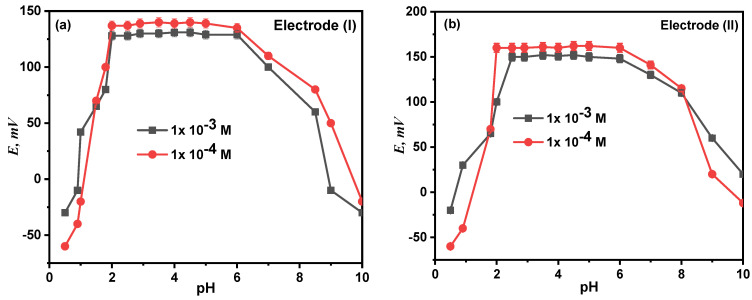
Effect of pH on the performance of proposed electrode (I) (**a**) and electrode (II) (**b**).

**Figure 9 molecules-28-08149-f009:**
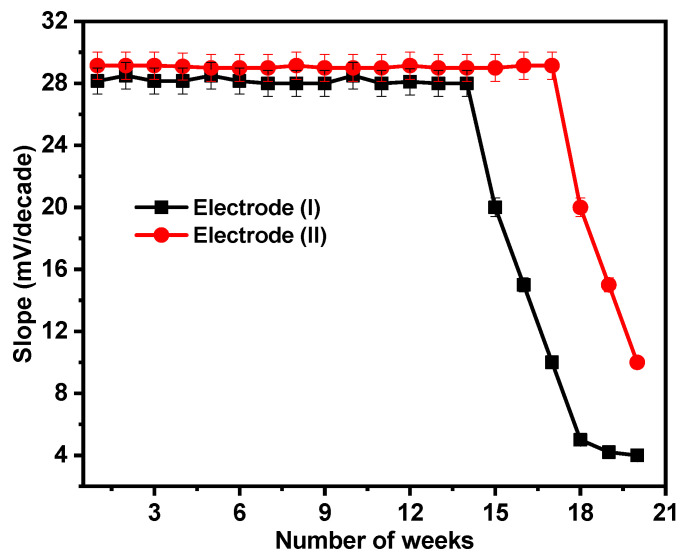
Reusability test for electrodes (I) and (II).

**Figure 10 molecules-28-08149-f010:**
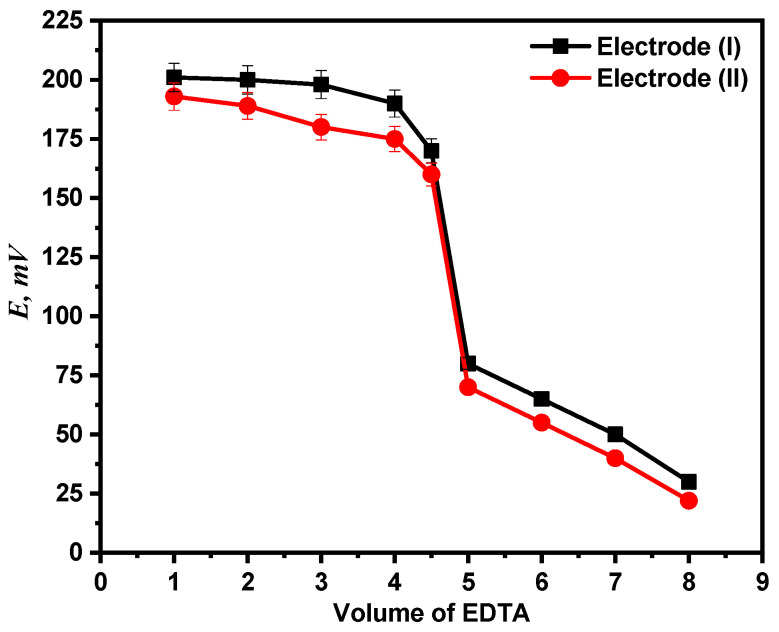
Potentiometric titration curve of Ca^2+^ ion solution with EDTA.

**Table 1 molecules-28-08149-t001:** Selectivity coefficient of proposed electrodes using SSM and MPM methods.

Series No.	Interfering Ions	*Log*[*K_A,B_*] SSM		*Log*[*K_A,B_*] MPM	
		Electrode (I)	Electrode (II)	Electrode (I)	Electrode (II)
1	Na^+^	−5.33	−6.33	−4.99	−2.33
2	K^+^	−6.56	−6.34	−2.34	−4.56
3	Mg^2+^	−5.64	−5.44	−3.33	−2.30
4	Cr^2+^	−5.44	−4.67	−4.33	−2.93
5	Ni^2+^	−5.07	−5.34	−3.40	−3.12
6	Mn^2+^	−5.13	−5.45	−3.45	−4.13
7	Cu^2+^	−5.14	−5.33	−2.75	−3.22
8	Hg^2+^	−4.95	−6.33	−3.34	−5.60
9	Zn^2+^	−6.33	−6.44	−4.99	−4.70
10	Cd^2+^	−5.43	−5.34	−3.56	−2.83
11	Co^2+^	−4.78	−5.77	−3.45	−3.81
12	Pd^2+^	−5.33	−4.33	−3.46	−4.32
13	Cr^3+^	−4.23	−4.44	−4.86	−4.77
14	Al^3+^	−4.78	−4.26	−4.18	−4.53
15	Fe^3+^	−3.99	−3.54	−4.08	−3.95

**Table 2 molecules-28-08149-t002:** Determination of Ca^2+^ in powdered milk and CAL-MAG drug samples using MCPEs (electrodes (I) and (II)).

Sample	Statistical Parameters	Electrode (I)		
		Calibration Graphs	Direct Method	Standard Addition Method
Powder milk	N	5.00	5.00	5.00
	Mean recovery (%)	98.54	99.30	99.56
	RSD (%)	0.88	1.00	0.79
CAL-MAG	N	5.00	5.00	5.00
	Mean recovery (%)	99.65	100.2	98.25
	RSD (%)	0.77	0.99	0.45
Powder milk	N	5.00	5.00	5.00
	Mean recovery (%)	98.67	99.50	99.54
	RSD (%)	0.89	0.77	0.44
CAL-MAG	N	5.00	5.00	5.00
	Mean recovery (%)	99.06	99.37	99.67
	RSD (%)	0.55	0.49	0.95
HPLC [2]	Mean recovery (%)	97.98		
	RSD (%)	1.03		

F-test = 0.02–0.88 and *t*-test = 1.0–1.40 for *n* = 5. 95% confidence limit: tabulated t value = 2.571 and tabulated F value = 5.05.

**Table 3 molecules-28-08149-t003:** Response properties of the proposed electrodes (I) and (II).

Parameters	Electrode (I)	Electrode (II)
Slope (mV decade^−1^)	28.15 ± 0.47	29.15 ± 0.47
Concentration range (µM)	1–1000	0.1–1000
Correlation coefficient (r^2^)	0.990	0.999
Working pH range	2.00–8.50	2.00–8.50
Isothermal coefficient (mV/°C)	1.33	1.50
Response time (S)	7.60	5
Detection limit (µM)	3.33	0.333
Recovery%	98.35–100.0	99.30–99.90
RSD%	0.43–0.98	0.34–1.00
Quantification limit (µM)	1–1000	0.1–1000

**Table 4 molecules-28-08149-t004:** Comparison between performance of proposed electrodes and some reported Ca^2+^-selective ion electrodes.

Reference	Slope (*mV*/Decade)	Linearity (mol/L)	SD	LOD (mol/L)	Response Time (s)
Proposed electrodes (I) and (II)	28.15 ± 0.47 and 29.15 ± 0.47	1.0 × 10^−6^–1.0 × 10^−3^ and 1.0 × 10^−7^–1.0 × 10^−2^	0.22–0.34 and 0.24–0.33	3.33 × 10^−6^ and 3.33 × 10^−7^	7.60 and 5
[33]	29.80	1 × 10^−5^–1 × 10^−1^	-	4.0 × 10^−6^	10
[34]	28.0 ± 0.2	1 × 10^−5^–1 × 10^−1^		4.0 × 10^−6^	30
[35]	28.00	2 × 10^−7^–1 × 10^−1^	0.45–0.55	2.0 × 10^−8^	15
[36]	-	2 × 10^−1^–4 × 1 × 10^−6^	-	-	15 to 120
[37]	30	1.00–1 × 10^−4^		1.0 × 10^−4^	60

## Data Availability

Data are contained within the article and Appendix A.

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
