# Peer review of "Graphitic Carbon Nitride/MOFs Hybrid Composite as Highly Selective and Sensitive Electrodes for Calcium Ion Detection"

_molecules, 2023, doi:10.3390/molecules28248149_

Round 1
Reviewer 1 Report
Comments and Suggestions for Authors
1. At certain places language is awkward. So, please check the minor inconsistencies.
2. Please illustrate the mechanism in detail.
3. Please check the morphology after Reusability test.
4. The Reusability test could be measured at least 4 recycles.
5. The subscript from the Fig3 could be revised.
6. Some related work on MOFs could be cited, such as Chemosphere, 2022, 307,135729 and New J. Chem., 2022, 46, 13818–13837
7. The PXRD is badly, I cannot find the main peaks from it, it could be checked the simulated mode
Comments on the Quality of English Languagework
Author Response
Reveiwer1: Comments and Suggestions for Authors
- At certain places language is awkward. So, please check the minor inconsistencies.
Answer: Thanks for the comment. We checked the language and grammar of our manuscript.
- Please illustrate the mechanism in detail.
Answer: Thanks for the suggestion. We illustrate the mechanism in detail as presented in page 7 line 248.
- Please check the morphology after the Reusability test.
Answer: Thanks for the suggestion. We did the morphology of samples for electrodes I and II (a, b) before soaking and (c, d) after soaking in 10−3 mol L−1 of Ca2+ ions as presented in Figure 5 page 8.
- The Reusability test could measure at least 4 recycles.
Answer: Thanks for the suggestion. We measured the reusability/or lifetime after storage the electrodes in the fridge in the interval (1-2-3 weeks).
- The subscript from Fig3 could be revised.
Answer: Thanks for the comment. We modified the Figure according to your observation as presented in page 6.
- Some related work on MOFs could be cited, such as Chemosphere, 2022, 307,135729 and New J. Chem., 2022, 46, 13818–13837
Answer: Thanks for the suggestion. We cited this work and other related work in the manuscript.
- The PXRD is badly, I cannot find the main peaks from it, it could be checked the simulated mode
Answer: Thanks for the comment. We modified the XRD Figure by adding the simulated XRD of MIL-125(Ti) as in page 5.

Reviewer 2 Report
Comments and Suggestions for Authors
The overall novelty is fari and may not be able to draw broad intentions from readers, but it proivde some advices for the MOF based materials in selective applications. Major comments
1. the novelty or statement of invention is required to be better and sharper
2. The scientific methods to illustrate the experiments should be enhanced
3. Systematic comparision with other similar references should be conducted to show its readability and clearance
4. Mechanism explanination is not clear, some statement is overestimated with strong supports. please enhance it.
Some specific comments
1. materials synthesis in section 2.2, 2.3, 2.4, 2.5, and 2.6, is this the novel method developed by Author? If not, at least a proper reference is compulsory to avoid any misleading. Proper citation in the right place is professionally required to meet research integrity criteria.
2. section 3.1 and XRD pattern in Figure 1, minimum SCDPF database or standard curve of the mentioned material is required to be the baseline. Or the typical peaks should be marked accordingly.
3. Figure 2. Scale bar with unit should be added to be professional
4. Figure 3. FTIR, the typical peak location should be marked accordingly to standard curves or other FTIR spectrum studies. Precise and correct peak location is required
5. Figure 4, a high resolution illustration with clear indication to show the pathway is required
6. Figure 5, what is the Y-axis unit? Slope? How to define it? standard error is required,
7. Figure 6 and 7 and 8 AND Figure 9 . Without standard error, hard to show the performance good or bad
8. What is the difference between Figure 5 and Figure 8?
9 Mechanism explanation is confused and not well supported.
Comments on the Quality of English LanguageWriting polishment is recommended.
Author Response
Reveiwer2: Comments and Suggestions for Authors
The overall novelty is fari and may not be able to draw broad intentions from readers, but it provides some advice for the MOF based materials in selective applications. Major comments
- the novelty or statement of invention is required to be better and sharper
Answer: Thanks for the comment. We improved the content of the manuscript to display the novelty.
- The scientific methods to illustrate the experiments should be enhanced
Answer: Thanks for the comment. We enhanced the method to become clear as in the experiment section in page 3.
- Systematic comparison with other similar references should be conducted to show its readability and clearance
Answer: Thanks for the comment. We added this comparison in the manuscript as presented in Table 4 page 14.
- Mechanism explanation is not clear; some statement is overestimated with strong supports. Please enhance it.
Answer: Thanks for the suggestion. We illustrate the mechanism in detail as presented in page 7 line 257.
Some specific comments
- materials synthesis in section 2.2, 2.3, 2.4, 2.5, and 2.6, is this the novel method developed by Author? If not, at least a proper reference is compulsory to avoid any misleading. Proper citation in the right place is professionally required to meet research integrity criteria.
Answer: Thanks for the comment. Some of it from literature and we added the proper citation of it (Materials Letters, 186 (2017) 151-154, Beilstein Journal of Organic Chemistry, 14 (2018), 1806-1812.), and the others are considered novel methods as the preparation of g-C3N4@Ti-MOF composite.
- section 3.1 and XRD pattern in Figure 1, minimum SCDPF database or standard curve of the mentioned material is required to be the baseline. Or the typical peaks should be marked accordingly.
Answer: Thanks for the comment. We modified the XRD Figure by adding the simulated XRD of MIL-125(Ti) as in page 5.
- Figure 2. Scale bar with unit should be added to be professional.
Answer: Thanks for the comment. We added the scale bar as presented in Figure 2 page 5.
- Figure 3. FTIR, the typical peak location should be marked accordingly to standard curves or other FTIR spectrum studies. Precise and correct peak location is required.
Answer: Thanks for the comment. We added the exact location of each peak in FTIR spectra.
- Figure 4, a high-resolution illustration with clear indication to show the pathway is required.
Answer: Thanks for the comment. We clarified the figure and explained it as shown in page 7 line 257.
- Figure 5, what is the Y-axis unit? Slope? How to define it? standard error is required.
Answer: Thanks for the comment. We added the unit of the slope in all Figures and, we clarified the standard error in our work in the Tables as exhibited in Table 3, 4.
- Figure 6 and 7 and 8 AND Figure 9. Without standard error, hard to show the performance good or bad.
Answer: Thanks for the comment. We added the standard error in all Figures.
- What is the difference between Figure 5 and Figure 8
Answer: Thanks for the comment. Figure 5 shows the behavior of electrodes with changing the temperature and Figure 8 the lifetime of the electrodes. In Figure 5 the x axis is the temperature but in Figure5 8 is the number of weeks.
- Mechanism explanation is confused and not well supported.
Answer: Thanks for the suggestion. We illustrate the mechanism in detail as presented in page 7 line 257 and supported the mechanism by doing the SEM-EDX for electrodes before and after soaking in Ca2+ solution as shown in Figure 5.

Round 2
Reviewer 2 Report
Comments and Suggestions for Authors
Revised manuscript basically address the comments from reviewers
Comments on the Quality of English LanguagePrefer to have a polishing to make it better for reading